# Effect of Two- and Three-Dimensionally Designed Guide Vanes with Different Camber Length on Static Pressure Recovery of a Wall-Mounted Axial Fan

Yong-In Kim [1,2], Yong-Uk Choi [3], Cherl-Young Jeong [3], Kyoung-Yong Lee [2] and Young-Seok Choi [1,2,*]

1    Industrial Technology (Green Process and Energy System Engineering), Korea University of Science and Technology, 217 Gajeong-ro, Yuseong-gu, Daejeon 34113, Korea; yikim89@kitech.re.kr
2    Clean Energy R&D Department, Korea Institute of Industrial Technology, 89 Yangdaegiro-gil, Ipjang-myeon, Seobuk-gu, Cheonan-si 31056, Chungcheongnam-do, Korea; chrisst@kitech.re.kr
3    Research Department, Geum-seong Fan Co., Ltd., 13 Sabaegobeon-gil, Asanbaelli-ro, Dunpo-myeon, Asan-si 31408, Chungcheongnam-do, Korea; dyddnr1335@naver.com (Y.-U.C.); heavyfan@naver.com (C.-Y.J.)
*    Correspondence: yschoi@kitech.re.kr; Tel.: +82-41-589-8337

**Abstract:** This study was based on a numerical effort to use the motor support (prop) as a guide vane when the motor of a wall-mounted axial fan was located at the fan outlet while maintaining the structural and spatial advantage. The design for the guide vane followed two- and three-dimensional methods. The inlet vane angle, meridional length (total), and meridional length with a vane angle of zero (0) degrees (linear) were considered as design variables. At the design and some low flow rate points, the 2D design offered the most favorable performance when the meridional length with a vane angle of zero (0) degrees (linear) was 30% based on total length, and was the worst for 70%. The 3D design method applied in this study did not outperform the 2D design. In the 2D design concept, averaging the flow angle for the entire span at the design flow rate could ensure a better pressure rise over a more comprehensive flow rate range than weighting the flow angle for a specific span. In addition, the numerical results were validated through an experimental test, with an important discussion of the swirl (dynamic pressure) component. The influence of the inlet motor and turbulence model are presented as a previous confirmation.

**Keywords:** axial fan; guide vane; swirl; slip; static pressure recovery; two-dimensional design; three-dimensional design; shear stress transport reattachment modification (SST R.M.)

## 1. Introduction

A wall-mounted axial fan aims for structural simplicity with no duct corresponding to the extended inlet and outlet flow passage to achieve a spatial advantage. Accordingly, a blade including a hub, an electric motor, and a shroud casing having a short axial (meridional) length constitute an assembly as the main components [1,2]. A bell-mouth and hub-cap are minimized, and guide vanes at the rear of the blade are generally omitted [3]. Figure 1 shows the typical assembly of a wall-mounted axial fan. Assuming the flow follows the black dotted arrow, the motor is placed at the fan inlet. In order to avoid electrical problems caused by outdoor environments such as weather at industrial sites, a wall-mounted axial fan is mainly installed so that the motor faces indoors to operate as the air exhaust. Here, the support (prop) for the motor should not have directionality (angle) in order not to affect the flow characteristics at the blade inlet according to theoretical considerations [4,5]. Of course, as the outdoor environment is ignored, the supply of air is also possible with a reverse installation. On the other hand, if the motor is located at the fan outlet as indicated by the black full arrow, it can be operated for the air supply (or exhaust) with (or without) consideration of the outdoor environment. In this case, interestingly, the support has a frame that can act as a guide vane at the rear of the blade; i.e., when a predetermined area and angle can be given to the motor support of the wall-mounted axial

fan having a motor at the blade outlet, the support may be understood as a guide vane of the wall-mounted axial fan.

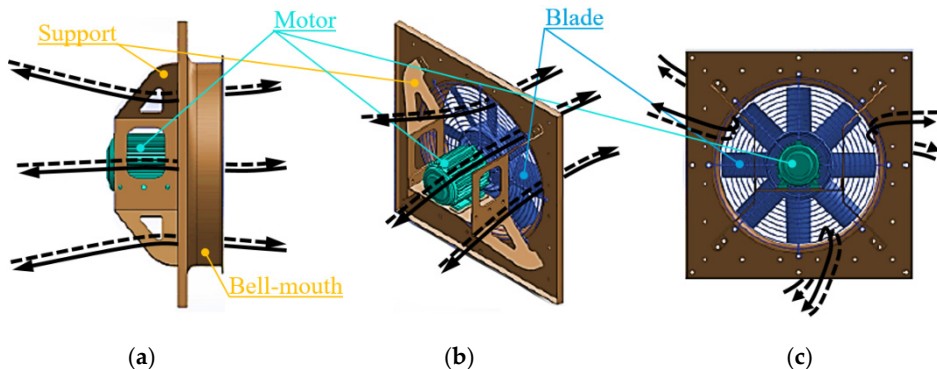

**Figure 1.** Typical assembly of a wall-mounted axial fan: (**a**) side view; (**b**) three-dimensional (3D) view; (**c**) front view.

If an axial fan goes beyond the wall-mounted type, such as fans applied to underground spaces and tunnels, the attachment of guide vanes is quite common, because a fan can include a duct with an appropriate length to the inlet and outlet [6–8]. The guide vane can improve the performance of an axial fan as a rear part of the blade. A vigorous swirl along the circumferential direction should be generated after the blade, and the guide vane forcibly controls the swirl (dynamic pressure) to restore static pressure. The role of the guide vane is essential as the swirl becomes stronger. The strong swirl contained at the blade outlet means that there is sufficient margin for static pressure recovery [9,10]; therefore, the design of the guide vane is worthwhile for the wall-mounted axial fan. The guide vane can be designed from the empirical selection of the effective parameters obtained through the analysis of flow patterns at the blade outlet [11]. However, if the outlet part is excessively extended, its structural and spatial advantages will be lost. Thus, it is necessary to impose a limiting condition on the axial length of the guide vane. Moreover, if the static pressure recovery is insufficient due to the limited axial length of the guide vane, the design method should be reviewed from various aspects.

This study was a numerical attempt for the case in which the guide vane has a short axial (meridional) length due to the limited space at the blade outlet, such as the wall-mounted type. As a general premise, it was assumed that the motor has a diameter equal to or smaller than that of the blade hub so that the motor is provided inside the hub at the blade outlet, so as not to affect the aerodynamic characteristics. The design method for the guide vane followed these two concepts: two-dimensional (2D) geometry exhibiting advantages in terms of mass production and cost of its manufacturing process with the plate rolled up at once; and three-dimensional (3D) geometry exhibiting advantages in terms of aerodynamic flow patterns with different inlet vane angles at each span. Figure 2 shows the schematic view for each. From the potential degradation expected from the limited axial length of the guide vane, the Results and Discussion section presents the pressure rise and internal flow field to the streamwise (axial) position of the guide vane. In addition, further design was carried out for the axial length and inlet vane angle of the guide vane. The description was mainly focused on the design flow rate point, including low flow rate points where flow separation became progressively more severe as the incidence and deviation angles increase. Meanwhile, the numerical results were validated through an experimental test, and here, a consideration of the swirl (dynamic pressure) component is discussed, with a deviation between the simulation and experimental test.

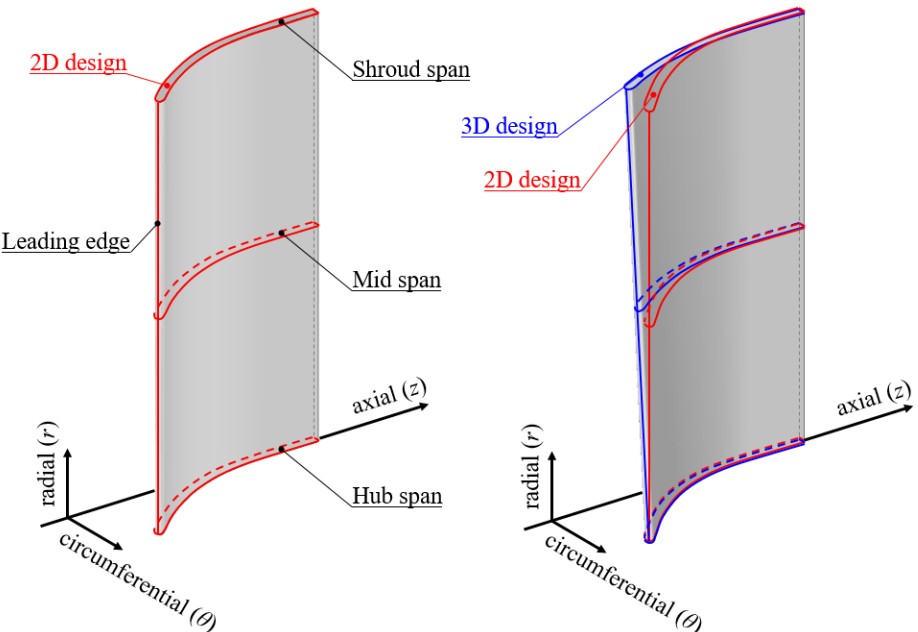

**Figure 2.** Schematic view of designed guide vane with 2D (red) and 3D (blue) geometry.

## 2. Axial Fan and Guide Vanes

### 2.1. Wall-Mounted Axial Fan Unit

The wall-mounted axial fan in this study exhibited the design specifications as listed in Table 1, where $D_h$ and $D_s$ denote the diameter of the hub and shroud. Detailed parameters of the blade are shown in Table 2. The blade had a shorter chord and meridional length at the shroud span than the hub span. The specific speed in type number ($N_s$), flow coefficient ($\Phi$), and pressure coefficient ($C_p$), were non-dimensionalized as follows:

$$N_s = \frac{\omega \sqrt{Q}}{(P_t/\rho)^{0.75}} \tag{1}$$

$$\Phi = \frac{C_{m2}}{U_2} \tag{2}$$

$$C_p = \frac{P_t}{0.5\rho U_2^2} \tag{3}$$

where $\omega$, $Q$, $P_t$, $\rho$, $C_{m2}$, and $U_2$ denote the angular velocity, volume flow rate, total pressure, air density, meridional component of absolute velocity at the blade outlet, and rotational velocity at the blade outlet, respectively. Here, the area at the blade outlet was considered as full area ($\pi D_s^2/4$), while the numerator in pressure coefficient ($C_p$) was substituted as static pressure ($P_s$) only in Section 5 (Results and Discussion).

**Table 1.** Design specifications of the wall-mounted axial fan.

| Specification | Value | Unit |
|---|---|---|
| Specific speed ($N_s$) | 3.9 | (-) |
| Flow coefficient ($\Phi$) | 0.19 | (-) |
| Pressure coefficient ($C_p$) | 0.24 | (-) |
| Rotational speed ($N$) | 880 | (rpm) |
| Hub ratio ($D_h/D_s$) | 0.41 | (-) |

**Table 2.** Design parameters of the blade.

| Parameter | Value | Unit |
|---|---|---|
| Chord length | 192.3 (hub), 178.3 (shroud) | (mm) |
| Meridional length | 138.8 (hub), 50.6 (shroud) | (mm) |
| Max. thickness (airfoil) | 12.9 (hub), 9.3 (shroud) | (mm) |
| Setting angle [1] | 46.2 (hub), 16.4 (shroud) | (degrees) |
| No. of blades ($Z_b$) | 10 | (-) |

[1] Based on tangential definition.

### 2.2. Guide Vanes

The main parameters for the guide vanes are listed in Table 3, except for the inlet vane angle, which is further described in the next paragraph. For the meridional length (total), a normalized value of 0.15 based on the axial fan's shroud diameter ($D_s$) was designated to maintain the advantages in spatial compactness; it tended to be as short as possible—shorter than the blade chord length of the hub and shroud, but slightly longer than the blade meridional length of the hub. The meridional length (linear) with the vane angle parallel to the axial direction (0°) was examined with normalized values of 0.1, 0.3, 0.5, and 0.7, respectively, based on the total length (150 mm). It was previously reported that the static pressure recovery showed better performance with the linear length shorter than the normalized value of 0.2 [11], but this was reviewed again for this study while considering the limited total length. On the hub span, the axial gap between the blade trailing edge and the vane leading edge was also limited to a normalized value of 0.05. The leading and trailing edges of the vane were treated as round (semi-circle) and square (cut-off) shapes. Figure 3 presents a side view when the guide vanes were attached as a rear part of the blade. The meridional plane of the guide vane had a rectangular shape. The extension between planes 3 and 4 is discussed in Section 5.2 (Further Design).

**Table 3.** Design parameters of the guide vanes.

| Parameter | Normalized Value (Unit) [1] | Actual Value (Unit) |
|---|---|---|
| Meridional length (total) | 0.15 (-) | 150 (mm) |
| Meridional length (linear) | 0.1, 0.3, 0.5, 0.7 (-) [2] | - |
| Axial gap [3] | 0.05 (-) | 50 (mm) |
| Vane thickness | 0.0045 (-) | 4.5 (mm) |
| No. of vanes ($Z_v$) | - | 11 (-) |

[1] Based on fan shroud diameter ($D_s$). [2] Based on actual value of meridional length (total) of 150 mm. [3] Between the blade TE and guide vane LE on the hub span.

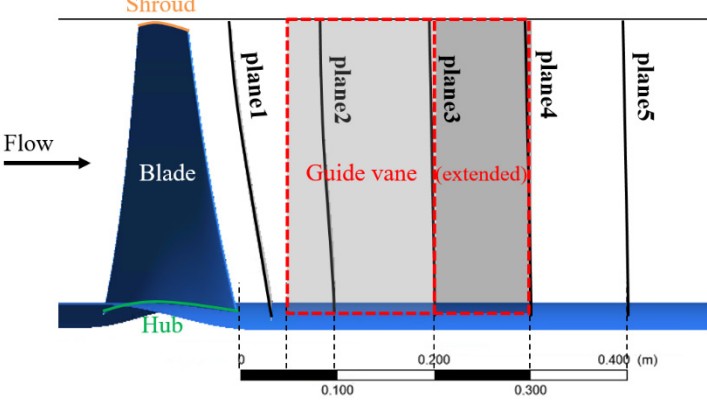

**Figure 3.** Side view of the blade and guide vane of the wall-mounted axial fan.

The inlet vane angle ($\beta_{1v}$) was selected from the absolute flow angle ($\alpha_2$) at the blade outlet. Therefore, a numerical analysis only with the blade should be conducted first

to obtain the actual flow pattern at the blade outlet; this was performed in a previous study [12] under the same numerical setup as this study. Figure 4a shows a schematic drawing of the velocity triangle at the blade outlet; the actual case was analyzed, not the ideal one. The absolute flow angle distribution over the entire span is shown in Figure 4b as circumferentially averaged data on planes 1 and 2 where the leading edge of the guide vane was located (see Figure 3). Here, the analysis was focused on the design flow rate ($1.0\Phi_d$). The 2D design, which should have the same vane angle for the entire span, progressed using 43.7° from the averaged data within the entire span at planes 1 and 2. The 3D design, which might secure more details for the flow patterns as 50–30° from the hub to the shroud span, was composed of approximated data within the specific span ($r^* \approx 0.1$–0.8) that could be distinguished as the mainstream in planes 1 and 2. The angle indicated as 2D′ (53.1°) was an additional consideration to give more weight to the flow patterns near the hub and shroud span (see Section 5.2). Each selected inlet vane angle should induce an incidence angle ($i_v$; $\alpha_2 - \beta_{1v}$) as shown in Figure 4c, where $\alpha_2$ denotes the averaged data of planes 1 and 2 for each $r^*$. Accordingly, the 3D design had almost no incidence angle within $r^* \approx 0.1$–0.8, but the 2D and 2D′ designs had both positive and negative values repeatedly. The maximum and minimum values for the incidence angle within $r^* \approx 0.1$–0.8 are noted in Figure 4c by the dotted cross lines.

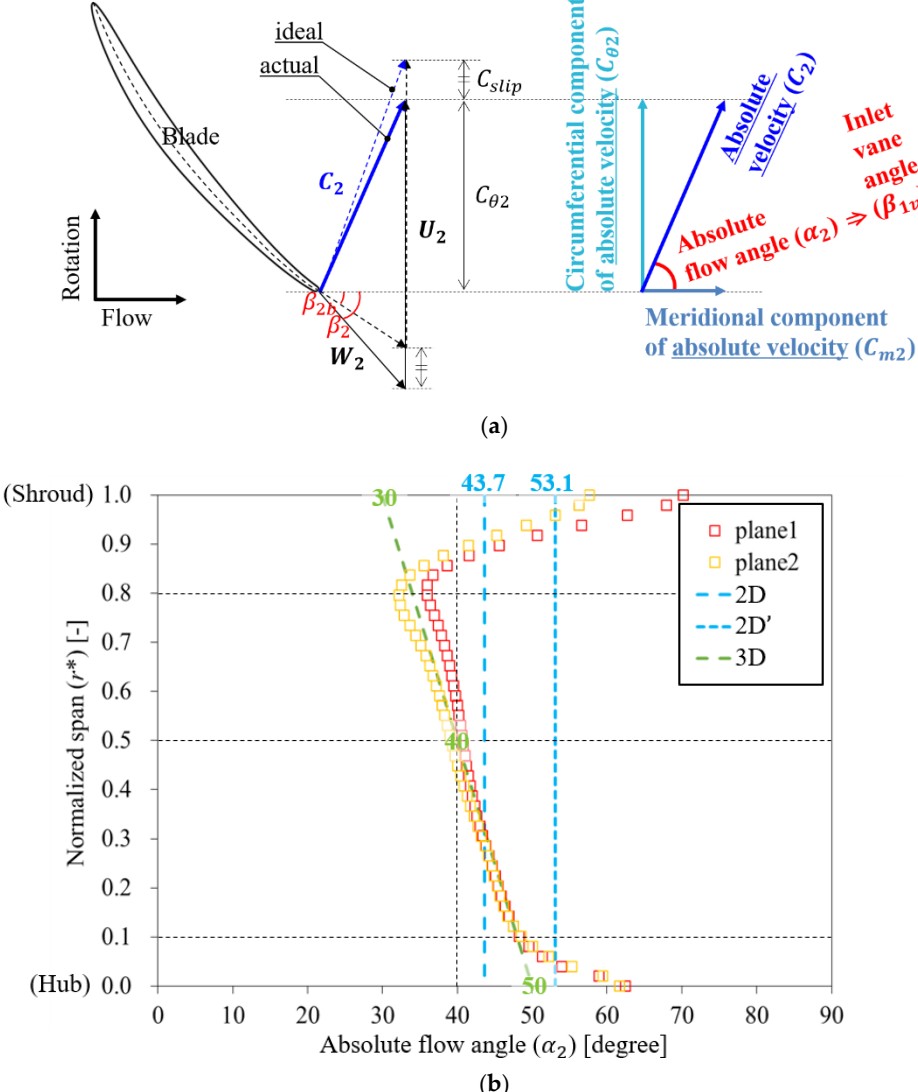

**Figure 4.** *Cont.*

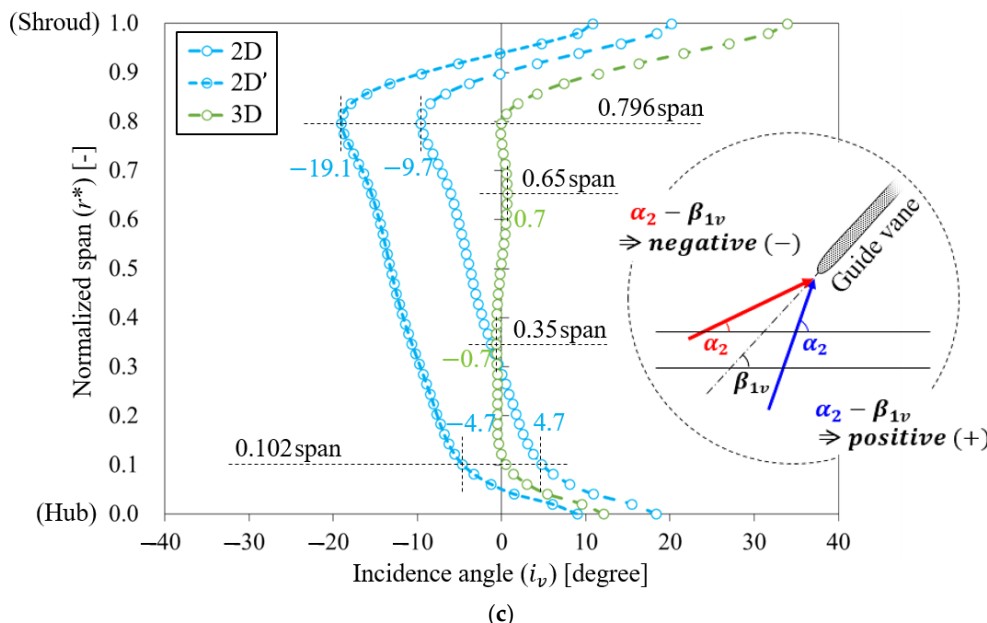

**Figure 4.** Selection for inlet vane angle (1.0 $\Phi_d$): (**a**) schematic drawing of velocity triangle at the blade outlet; (**b**) absolute flow angle distribution over the entire span; (**c**) incidence angle distribution over the entire span.

As a guideline for application to the other fans, the slip factor ($\sigma$) can be an index. From the ideal case indicated by the dotted triangle in Figure 4a, the relative velocity ($W_2$) should follow the outlet blade angle ($\beta_{2b}$); however, the slip must occur due to the deviation between the outlet blade angle and outlet flow angle ($\beta_2$). This study was based on the absolute flow angle ($\alpha_2$), including slip. Figure 5 shows the slip factor distribution over the entire span, which could be non-dimensionalized as follows:

$$\sigma = 1 - \left( C_{slip} / U_2 \right) \tag{4}$$

where $C_{slip}$ denotes the slip velocity, which was indicated in Figure 4a. In Figure 5, the proposed slip factors could be focused on the properties within $r^* \approx 0.1$–0.8, because the flow patterns near the hub ($r^* < 0.1$) and shroud ($r^* > 0.8$) could be subject to distortion, as shown in Figure 4b. Meanwhile, there was little change in the slip factor distribution from planes 2 to 5. The distribution that could be interpolated between the data of planes 1 and 2, where the inlet vane angle was selected, also did not show a significant difference from the distributions for the other planes. This meant that the axial gap between the blade and guide vane was difficult to be an effective variable. On the other hand, from the consideration of the incidence angle (Figure 4c) and slip factor (Figure 5) distribution, a rough design could be derived without previous analysis of the blade outlet flow pattern; however, since the slip factor was affected by design variables such as the number of blades, it can be used in cases with similar specifications to those of this study.

Finally, as shown in Figure 6, four sets of guide vanes were designed for each 2D and 3D geometry. Each inlet vane angle had the value above, and the vane angle ($\beta_v$) gradually became parallel to the axis ($0°$, $d\theta = 0$) from the leading edge under the following equation:

$$tan\beta_v = \frac{rd\theta}{dm} \tag{5}$$

where $r$, $\theta$, and $m$ denote the fan radius, tangential (circumferential) angular coordinate, and meridional length (total), respectively; this is depicted by the enlarged dotted circle in Figure 6a.

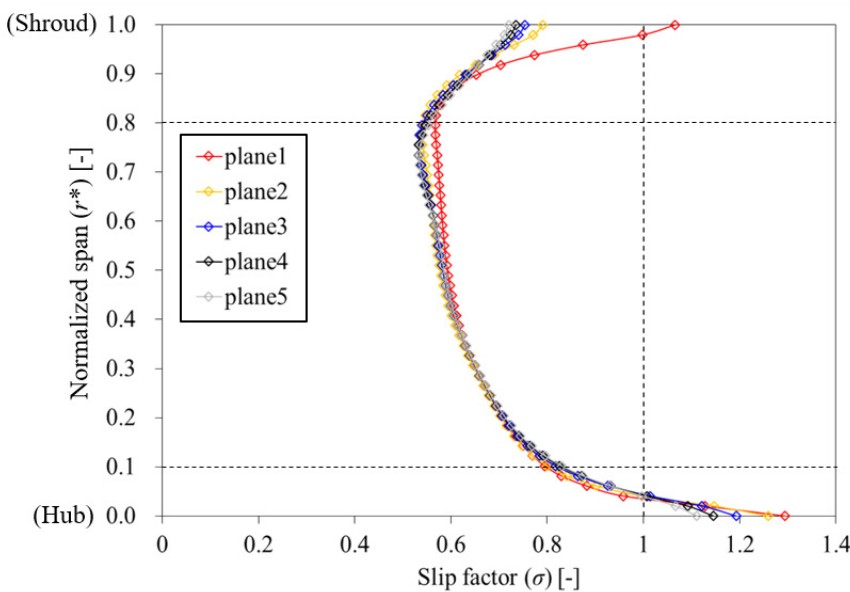

**Figure 5.** Slip factor distribution over the entire span for the wall-mounted axial fan (1.0 $\Phi_d$).

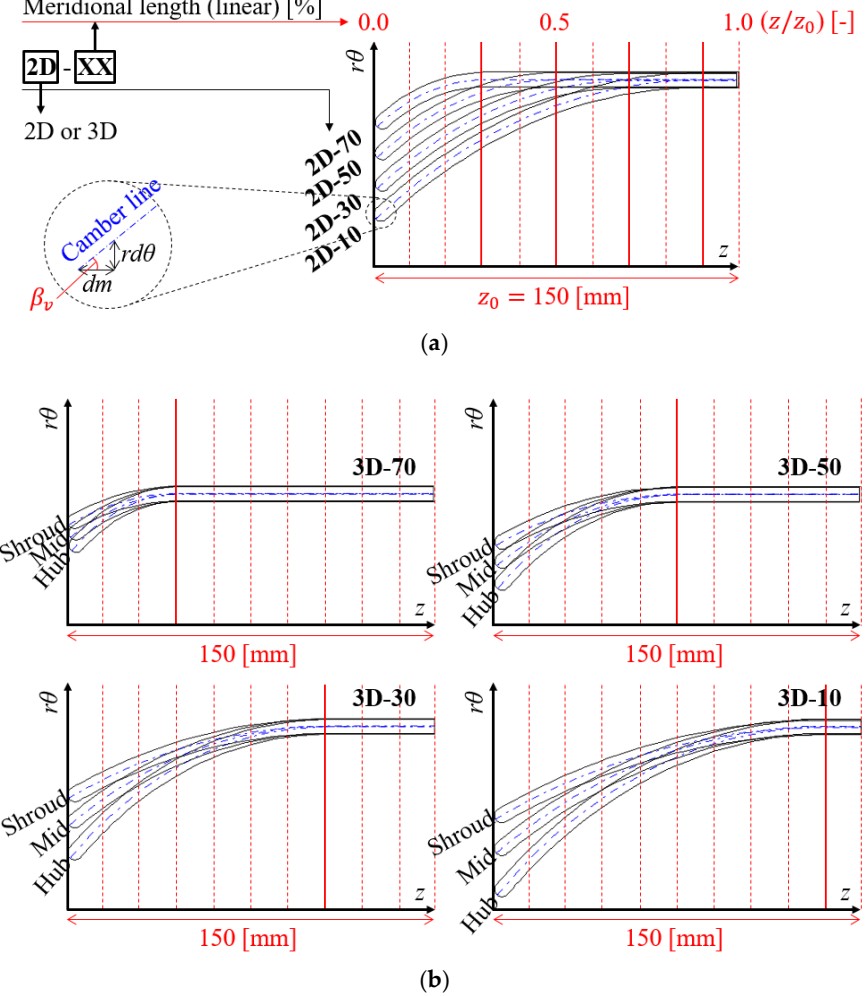

**Figure 6.** Designed guide vanes using (**a**) 2D and (**b**) 3D geometry in blade-to-blade view.

### 2.3. Inlet Motor Effect

Additionally, the influence of the inlet motor was numerically investigated as a previous step [12] to be examined before applying the outlet guide vane of a wall-mounted axial fan. In this study, the guide vane was applied as motor support at the rear of the blade, so it was desirable to examine the case in which the inlet motor was moved to the outlet. Since the examination tended to concentrate on the inlet motor regardless of the guide vane, the numerical domain was prepared only for the blade (rotor) without the guide vane (stator). The model for examination had slightly modified blades under the same design specifications and numerical setup as this study. Meanwhile, the space due to the removal of the inlet motor could be replaced with a hub-cap. Here, since an axial fan can exhibit almost the same aerodynamic performance as a straight flow passage if the inlet hub-cap has an approximate curvature, such as roughly round or elliptical shape [13], the geometry of the inlet hub-cap could be replaced with a straight flow passage. The outlet flow passage with jet-like flow patterns was also replaced with a straight flow passage.

The performance curves for the influence of the inlet motor were predicted (Figure 7). From each tendency, the inlet motor in this study had little effect on the performance of the axial fan, especially near the design flow rate ($\Phi \approx 0.19$, 1.0 $\Phi_{des}$). Figure 8 shows the internal flow patterns at the design flow rate. In Figure 8a, the 3D streamlines in a single passage contained no significant features for either the numerical domains of the inlet motor or straight flow passage. The motor seemed to act as a hub-cap at the blade inlet. As shown in Figure 8b, a very low velocity was formed in the space (cavity) between the motor and blade. Most of the mainstream flowed into the blade along a free stream at a height similar to the maximum diameter of the motor. Therefore, that space could be classified as a dead region; in other words, it was reasonable to remove the inlet motor and install a hub-cap.

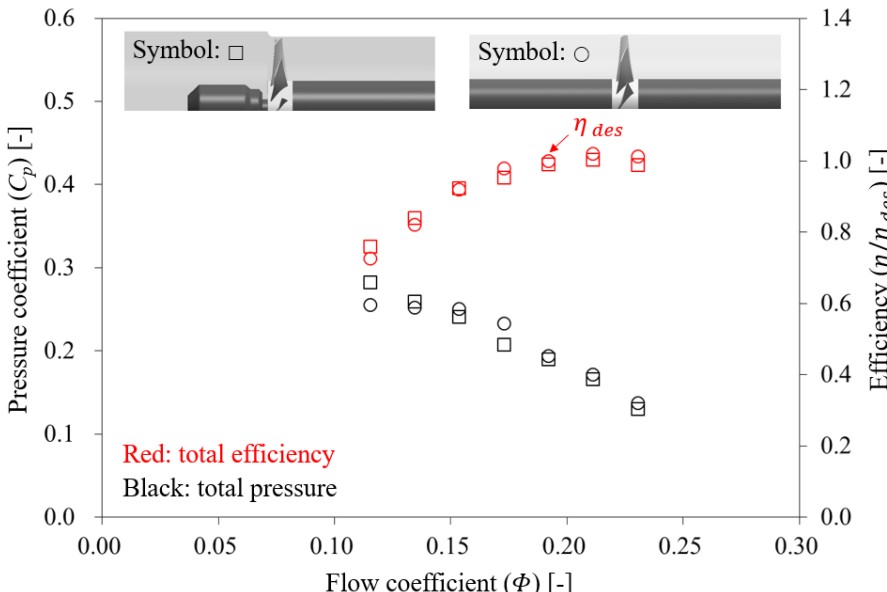

**Figure 7.** Performance curve for the influence of the inlet motor: square (inlet motor); circle (straight flow passage).

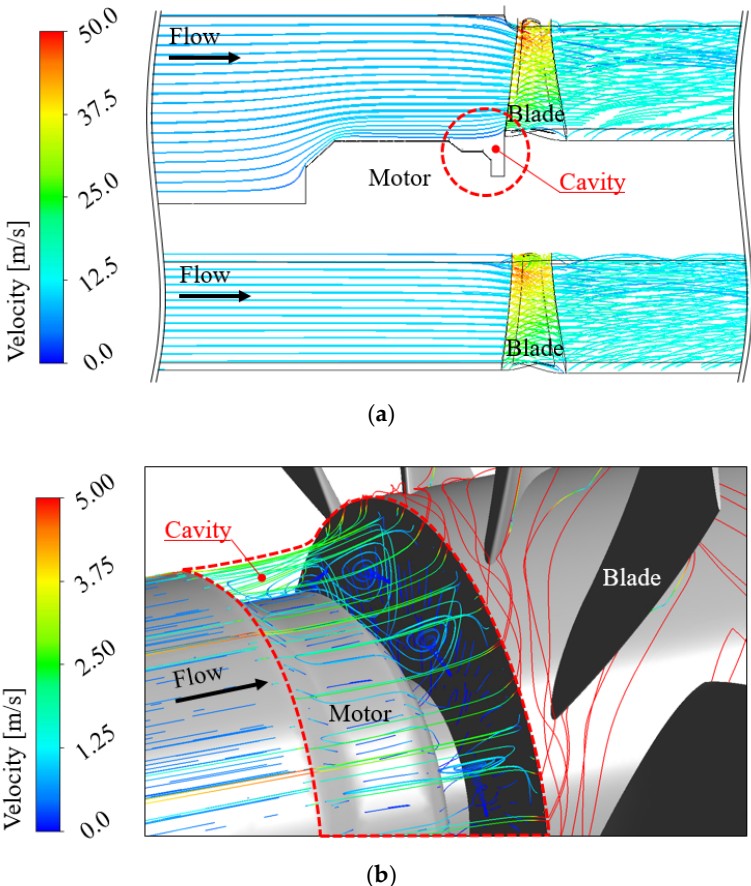

**Figure 8.** Internal flow pattern for the influence of the inlet motor (1.0 $\Phi_d$). (**a**) 3D streamlines in a single-passage for inlet motor (upper) and straight flow passage (lower); (**b**) 3D streamlines near the dead region.

## 3. Numerical Setup

### 3.1. Computational Domain and Grid System

Figure 9a shows the entire flow passage and grid system for numerical analysis. The computational domain consisted of both the rotor (blade) and stator (guide vane). The tip clearance was applied to the blade. The hub-cap (or motor) and bell-mouth that could be attached in front of the blade were not considered in the computational domain because they had almost the same effect as the extended straight passage [13], as mentioned in Section 2.1 (Wall-Mounted Axial Fan Unit). Any factors that did not significantly affect the performance of the axial fan could be excluded through preliminary confirmation, so that such an analysis enabled intensive care for the numerical convergence and the intrinsic impact of the object.

The grid system was generated using ANSYS CFX TurboGrid 19.1 as a hexahedral type. The grid system near the inlet surface and LE are shown in Figure 9a. The averaged $y^+$ on the blade and guide vane surfaces were 9.1 and 6.2, respectively. Under the same topology factor of the $y^+$, the grid test was conducted, as shown in Figure 9b. The grid system that did not affect the axial fan's performance was selected in response to the G3 set.

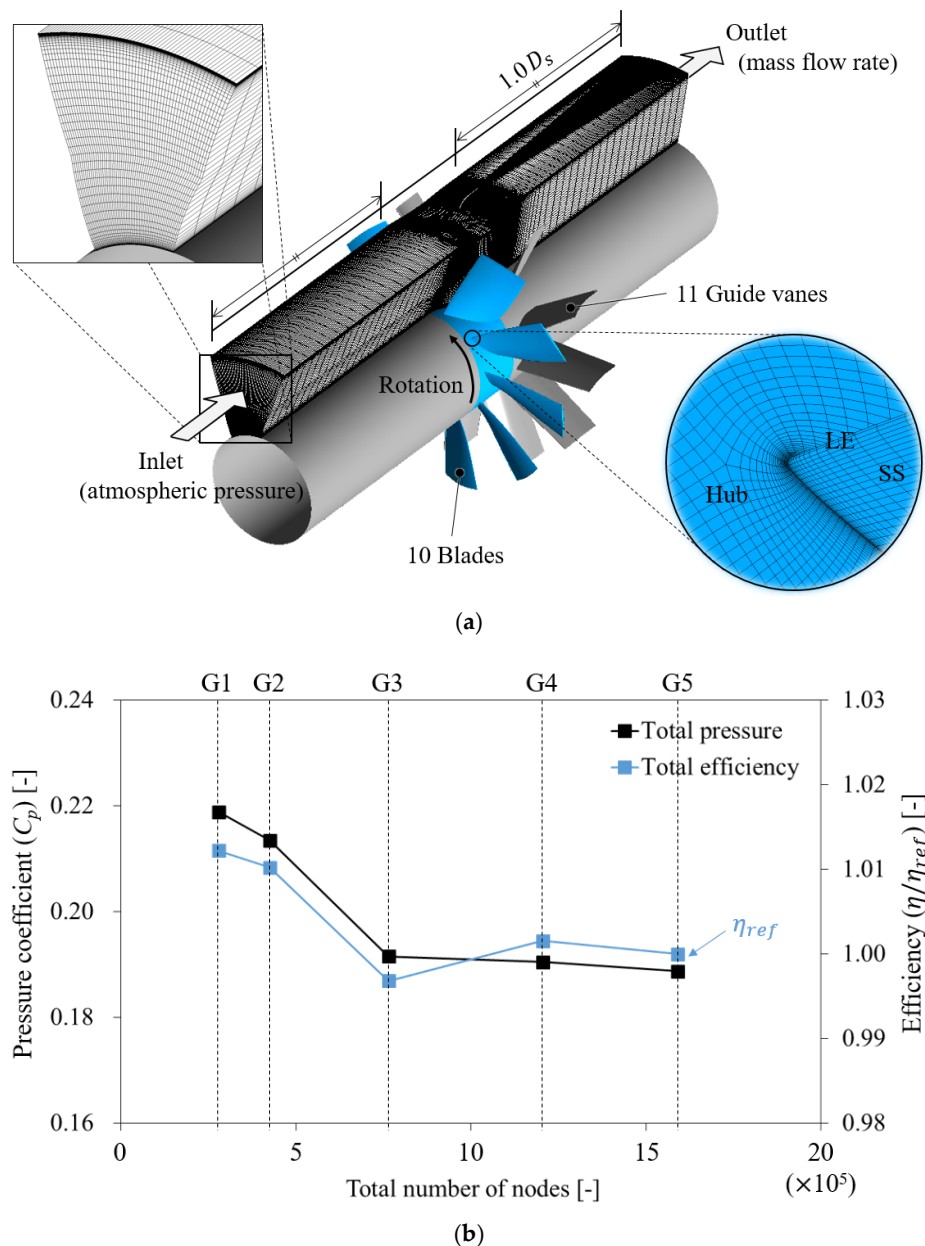

**Figure 9.** Computational domain and grid system: (**a**) flow passage; (**b**) grid test.

### 3.2. Governing Equation and Turbulence Model

The Reynolds-averaged Navier–Stokes (RANS) equations were applied to simulate the 3D steady-state analysis. The equations were based on the conservation laws for mass, momentum, and energy; however, the equation for the conservation of energy could be ignored so that the focus was on the isothermal (25 °C) condition. There was also no change in density over time because the maximum (local) Mach number near the shroud tip in the flow passage was 0.133 based on 25 °C, and could be regarded as a subsonic flow (Mach number < 0.3). In general, subsonic flow could be classified as an incompressible flow with no change in density. Meanwhile, a high-resolution discretization with a second-order approximation was applied to ensure the minimized numerical convergence instead of first-order schemes [14–16]. The root mean square (RMS) residuals of the governing equations for mass and momentum were kept below $1.0 \times 10^{-5}$.

In terms of the turbulence model, the $k$–$\omega$ based shear stress transport standard (SST Std.) model, which is widely known to be suitable for simulation of turbomachinery,

was developed to provide highly accurate predictions of the amount and onset of flow separation under adverse pressure gradients, including the transport effects into the eddy-viscosity formulation [17]. That is, the SST Std. model could exaggerate flow separation from smooth surfaces under the influence of adverse pressure gradients. Here, a modified SST Std. model, known as the shear stress transport reattachment modification (SST R.M.) model, may improve separation and reattachment predictions [18,19]. It modified the SST Std. model to enhance turbulence levels in the separating shear layers emanating from walls. Actually, the ratio of turbulence production to dissipation was greatly exceeded in regions of large flow separation from smooth surfaces, and the SST R.M. model obtained an additional source term in the $k$-equation to secure the ratio of turbulence production to dissipation [20]. On the other hand, as shown in Figure 10, there was little difference between the two models near the design point; that is, the application of each model had little effect on the state of weak separation. Instead, the unstable slope seemed to be improved at a specific flow rate ($\Phi \approx 0.17$, $0.9\, \Phi_{\text{des}}$). Finally, it was confirmed that any of the two models could be selected, and in this study, the SST R.M. model was applied. Here, the confirmation of the two turbulence models was performed with a numerical domain in which the inlet and outlet extended in a straight flow passage, as shown in Figure 9a.

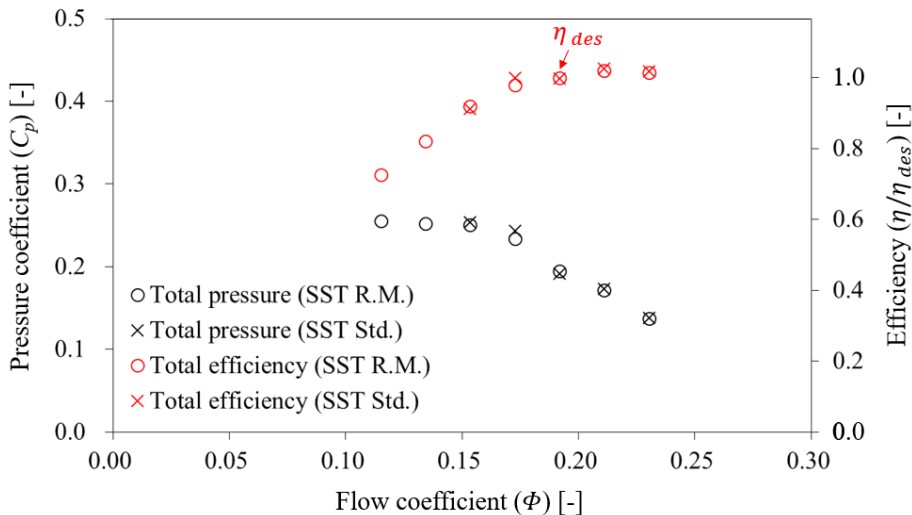

**Figure 10.** Performance curve for the influence of the turbulence models.

### 3.3. Boundary Condition

As also shown in Figure 9a, the periodic condition was given to the numerical domain so that the single passage with one blade and one guide vane could be selected. The stage (mixing-plane) method was applied to the interface between the rotor (blade) and stator (guide vane) because there was a stator as a rear part of the rotor. Although the aerodynamic variables were averaged in the rotational direction at the interface, it could suppress the influence of the relative frame between domains in the steady-state analysis with two or more domains. The wall boundaries were treated as a non-slip and smooth surface, and the automatic wall function was assigned to avoid the effect of the $y^+$ on the numerical results [21–23]. A strict low-Reynolds-number implementation of the $k$–$\omega$ model would also require a near-wall grid resolution of at least $y^+ < 2$; however, the automatic wall function allowed for a shift from a low-Reynolds-number form to a wall-function formulation. Meanwhile, the flow direction from the inlet was set as normal, and the options of atmospheric pressure and mass flow rate were given to the inlet and outlet boundaries, respectively. As the inlet boundary condition, the turbulence intensity ($T_u$) was set as a 'medium' level, which corresponded to a range of 1–5%. The medium turbulence intensity is recommended for flow in not-so-complex devices such as large pipes, fans, wind tunnels, or ventilation flows [18]. The actual turbulence intensity as determined by

the circumferentially averaged turbulent kinetic energy (TKE; $k$) at the inlet plane was about 4.3%.

$$T_u = \sqrt{\frac{\frac{1}{3}[(u - \overline{u})^2 + (v - \overline{v})^2 + (w - \overline{w})^2]}{(\overline{u}^2 + \overline{v}^2 + \overline{w}^2)}} = \sqrt{\frac{\frac{2}{3}k}{(\overline{u}^2 + \overline{v}^2 + \overline{w}^2)}} \tag{6a}$$

$$k = \frac{1}{2}[(u - \overline{u})^2 + (v - \overline{v})^2 + (w - \overline{w})^2] \tag{6b}$$

where $u$, $v$, $w$, $\overline{u}$, $\overline{v}$, and $\overline{w}$ denote the instantaneous velocity and the averaged velocity of $x$-, $y$-, and $z$-directions in the orthogonal coordinate system, respectively.

### 3.4. Solver Information

The commercial software ANSYS CFX 19.1 was used for numerical simulation. The parallel computations were conducted in PCs with Intel® Xeon® CPU E5-2680, clocked at 2.70 GHz with dual processors. The time spent per each set was approximately 1 h.

### 4. Experimental Validation

Before analyzing the numerical results, an experimental test was performed to validate the numerical method and results. The validation was conducted only with the blade (without the guide vane). In the numerical analysis, as shown in Figure 9a, the inlet and outlet of the fan had an extended domain. Assuming that the influence of other attachments (hub-cap, inlet motor, etc.) was previously confirmed, as in this study, the extended domain took numerical convergence as the most beneficial advantage. Additionally, the brevity of iterative analysis was strongly encouraged. It was not appropriate to perform the numerical analysis with complex and extensive geometries in the experimental test. Nevertheless, validation between the numerical analysis and experimental test was required to be conducted.

Figure 11a [24] shows the outlet chamber setup with flow settling means [25] and multi-nozzles among various methods, which were compliant with the international standard, ANSI/AMCA 210 [26]. This type of setup allowed the wall-mounted axial fan to be connected directly to the chamber, and was suggested as suitable when the outlet swirl is vigorous. Figure 11b shows a prototype of the wall-mounted axial fan. It included a hub-cap and bell-mouth at the fan inlet, and had the form of immediately open space at the blade outlet (free outlet) without additional attachments. Therefore, the swirl (dynamic pressure) component existing in the blade outlet could not be restored to the static pressure and mostly disappeared; i.e., the total pressure indicated in the experimental test could be the sum of the static pressure rise only from the rotating blades and the dynamic pressure corresponding to the fan outlet area. Accordingly, the total pressure obtained in the experimental test needed to be carefully compared with that of the numerical analysis. A straight flow passage extending as a rear part of the blade was applied in the numerical analysis. At the outlet of the straight flow passage, the flow pouring out from the blade was still maintained; i.e., the swirl (dynamic pressure) component at the blade outlet, which disappeared in the experimental test, could be obtained in the numerical analysis. Finally, the total pressure predicted in the numerical analysis should be the sum of the static pressure rise only from the rotating blades, the dynamic pressure corresponding to the fan outlet area, and the dynamic pressure of the swirl. Therefore, if no attachment can control the swirl at the blade outlet as in the experimental conditions of this study, the total pressure level in the numerical analysis must always be higher than the experimental test, and the stronger the swirl remains, the larger the deviation of the total pressure level. Eventually, this was confirmed as notable, as shown in Figure 11c.

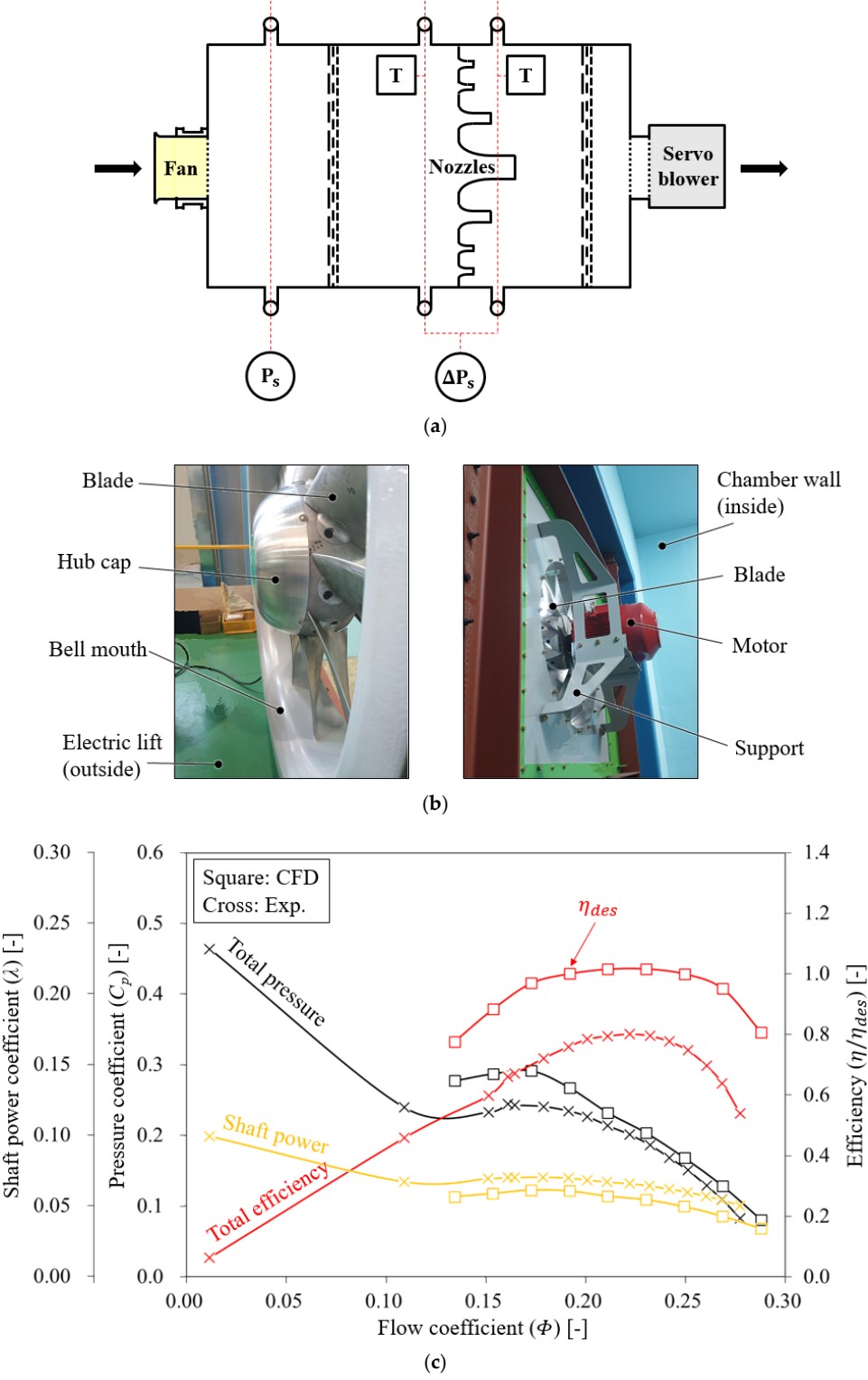

**Figure 11.** Experimental setup and results: (**a**) schematic diagram of the test facility; (**b**) prototype of the wall-mounted axial fan; (**c**) performance curve for the validation.

Figure 11c shows the mutual validation of the results obtained from the numerical method and the experimental test, as described above. Here, the total pressure difference between the inlet and the outlet was extracted to obtain the numerical results, and the sum of the static pressure and the dynamic pressure at the fan outlet was obtained for the experimental results in compliance with international standards [26]. The rotational

speed and air density during the experimental test were converted corresponding to the numerical condition. The shaft power of the experimental results was measured using a torque meter so that it could be directly compared with the numerical data. The shaft power coefficient ($\lambda$) was defined as follows:

$$\lambda = \frac{T\omega}{\frac{\rho}{2}A_2 U_2{}^3} \tag{7}$$

where $T$ and $A_2$ denote the torque for the impeller blade and hub, and the area of the blade outlet, respectively. From Figure 11c, it was confirmed that the tendencies between the numerical analysis and experimental test on the performance curve generally agreed well with each other. In the low flow rate range ($\Phi < 0.19$), the deviation of the total pressure between the experimental test and numerical analysis tended to gradually increase due to the swirl remaining inside the extended outlet passage in the numerical domain, as mentioned in the above paragraph. Nevertheless, the slope's inflection points on the total pressure and total efficiency curves, perceived as the stall's inception and best efficiency point (BEP), were quite accurate. The higher prediction for the total efficiency curve was due to the higher pressure and lower torque in the numerical analysis. The actual deviation of the torque was almost constant, about 400 W for the entire flow rate range. It could be confirmed as an ideal case, since the numerical results showed such aerodynamic affinity, rather than the experimental results, and it should be understood from the perspectives of mechanical loss and roughness.

If the numerical analysis follows all the geometric conditions of the experimental test, the total pressure can be extracted in the way suggested by the international standard [26]; i.e., the total pressure can be estimated as the sum of the static pressure and the dynamic pressure at the fan outlet. In this case, however, one more important point should be considered regarding the fan outlet area for the dynamic pressure calculation. If the fan outlet area is applied as the donut-shaped surface ($\pi r_s{}^2 - \pi r_h{}^2$), excluding the area corresponding to the hub diameter from the shroud diameter, a higher dynamic pressure will be substituted for the total pressure compared to the outlet area only considering the shroud diameter ($\pi r_s{}^2$). The international standard [26] generally applies dynamic pressure to the outlet area considering the shroud diameter ($\pi r_s{}^2$). In this study, however, the validation was properly performed by comparing the total pressure difference of the numerical analysis to the sum of the static and dynamic pressure of the experimental test.

## 5. Results and Discussion

### 5.1. Two- and Three-Dimensional Designs

In this section, the normalized meridional length (total) of the guide vane was 0.15 based on the fan shroud diameter ($D_s$). Figure 12 shows the static pressure rise (*y*-axis) at each axial distance (*x*-axis); the axial distance was normalized for the fan shroud diameter ($D_s$); the origin (0 $D_s$) was set to the leading edge of the guide vane. The black lines in a vertical direction mean the meridional length occupied by the blade hub (full-line) and shroud (dotted-line), and red lines in a vertical direction mean the meridional length corresponding to the guide vane. The black lines in a horizontal direction with the black arrows indicate the maximum level of static pressure recovery from the numerical analysis only with the blade. Assuming that the swirl (dynamic pressure) at the blade outlet was completely stabilized, the maximum level of static pressure recovery, which would be improved from the guide vanes, could be estimated for the wall-mounted axial fan. The gap between the numerical simulation only with the blade (none) and the maximum level of static pressure recovery became more prominent as the flow rate decreased, which meant that the swirl component also increased as the deviation angle increased. Meanwhile, the static pressure in the numerical analysis for this section may have no meaning as an actual value in contrast to the total pressure in the numerical analysis for the validation

section above, and it would be more appropriate to compare as a relative index among the designed sets.

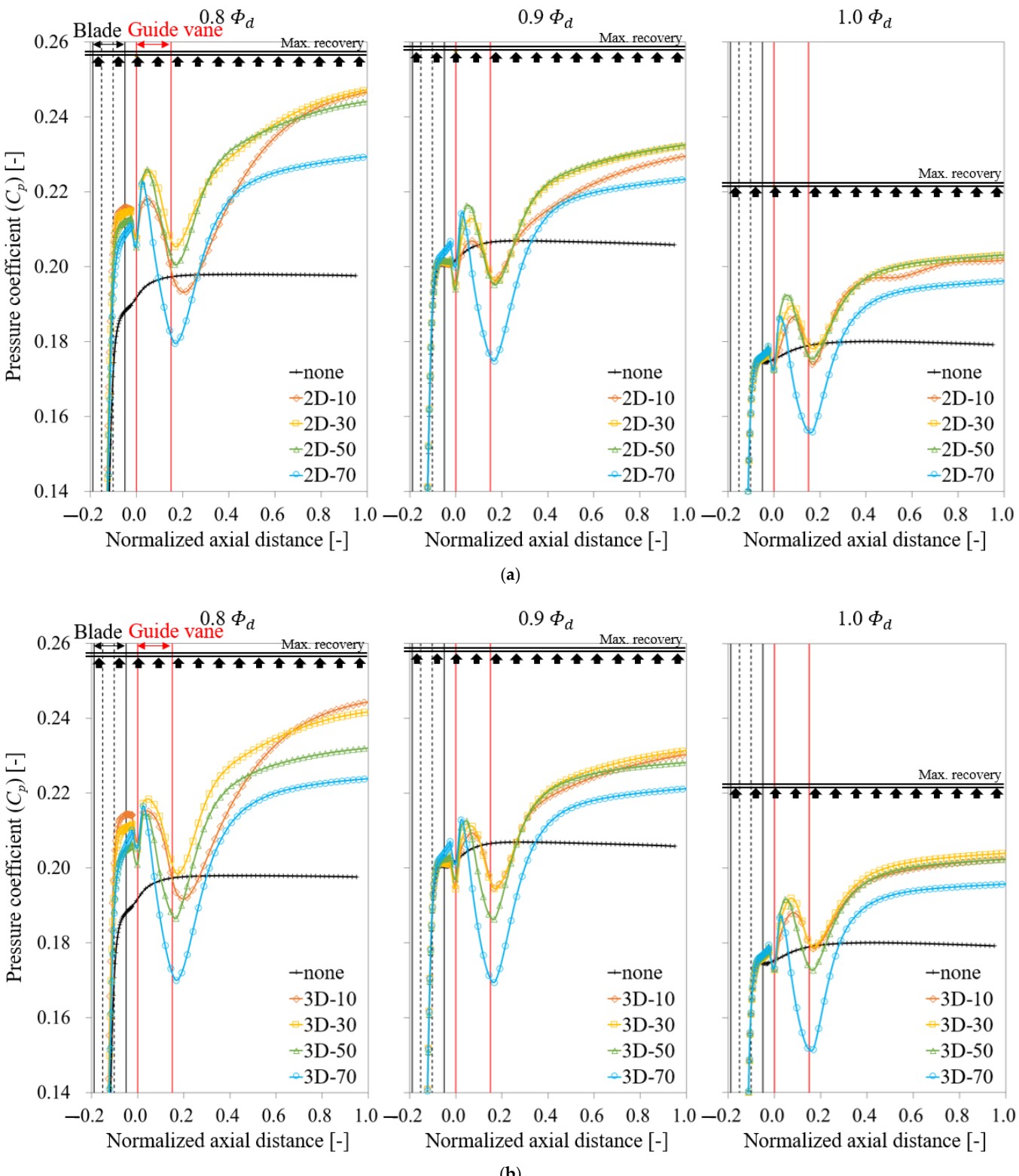

**Figure 12.** Static pressure rise in axial distance: (**a**) 2D design; (**b**) 3D design.

First, Figure 12a relates to the 2D design. At 1.0 $\Phi_d$ (design point), the static pressure rise, which showed almost the same pattern before passing through the blade, denoted

a difference as it encountered the guide vane. All four sets experienced static pressure recovery in different patterns, but lost the entire recovery at the trailing edge (end) of the guide vane. These were at an even lower level than without the guide vanes. The 2D-70 set contained the most noticeable degradation. The 2D-50 set seemed to have the best resilience in the guide vanes. However, it was nearly identical to the 2D-10 and 2D-30 sets, and was just temporary. Here, all four sets showed a drastic pressure rise from about 0.2–0.4 $D_s$. This point would be provided with a duct extending approximately 0.25 $D_s$ from the trailing edge of the guide vane. The 2D-70 set still had the lowest static pressure level. At 0.9 $\Phi_d$, the pressure rise pattern was almost similar to that of 1.0 $\Phi_d$, but the static pressure of the 2D-70 set was affected even before passing through the blade. At 0.8 $\Phi_d$, the static pressure of all sets was affected before passing through the blade. At the trailing edge of the guide vanes, the 2D-30 and 2D-50 sets were observed with a higher static pressure level than with no guide vanes. In 0.8–1.0 $\Phi_d$, the 2D-30 set proved to be the most advantageous, while the 2D-70 set obtained unfavorable characteristics. In addition, when the duct extended at about 0.65 $D_s$ from the trailing edge of the guide vane (about 0.8 $D_s$ point in the graph), the range of the static pressure recovery due to the attachment of the guide vane generally increased as the flow rate decreased.

Figure 12b relates to the 3D design. Each pattern showed a level of static pressure recovery almost similar to or slightly lower than that of 2D. Unfortunately, using the 3D geometry from the concept shown in Figure 4b, it was difficult to obtain any significant advantage over 2D. A 3D design that can more actively account for the flow characteristics near the hub or shroud spans may be expected to replace the 2D design. However, a 3D design will add complexity to the manufacturing process. Meanwhile, the pressure rise was dramatic in both the 2D and 3D designs when a duct was provided that extended by about 0.25 $D_s$ from the trailing edge of the guide vane (about 0.4 $D_s$ point in the graph).

The internal flow fields for sets of none, 2D-10, 2D-70, 3D-10, and 3D-70 are shown in Figure 13a–e, respectively. The black dotted lines in a vertical direction denote the ticks for the *x*-axis, which means the normalized axial distance based on the fan shroud diameter, the same as that in Figure 12. In the absence of guide vanes (Figure 13a, none), a fairly strong swirl was distributed in the outlet passage. The intensity of the swirl increased further as the flow rate decreased, which could be quantified as the difference of maximum recovery in static pressure, as indicated in Figure 12. On the other hand, as shown in Figure 13b, the guide vanes could act as a de-swirler. Here, an opposite effect was obtained near the hub span. The 2D-10 set did not fully account for the flow pattern near the hub span, causing severe separation. At 1.0 $\Phi_d$, the flow passage where the guide vane was placed almost seemed to be blocked, and at 0.8 $\Phi_d$, a back-flow was observed eventually. These unstable flow patterns near the hub span were almost stabilized in the mid-span, and were quite vane-friendly near the shroud span. If a design capable of anti-separation near the hub span was combined with the hub profile of the 2D-10 set, the static pressure rise curve in Figure 12a would be closer to the maximum recovery line. In Figure 13c, the streamline near the hub span for the 2D-70 set showed better patterns than the 2D-10 set. However, it contained separation over the entire span, including the mid- and shroud spans. This seemed to be the main cause of the worst static pressure recovery of the 2D-70 set. On the other hand, both the 2D-10 and 2D-70 sets generally resolved separation after about 0.4 $D_s$, which could have been the cause of the dramatic pressure drop and rise observed at about 0.1–0.4 $D_s$ as shown in Figure 12. Figure 13d,e relate to the 3D design, and each pattern was almost identical to those of the 2D design. However, the flow patterns near the shroud span for the 3D-10 and 3D-70 sets contained stronger separation than those of 2D. This result also corresponded to a slightly lower level of static pressure recovery in the 3D design than the 2D design, as shown in Figure 12.

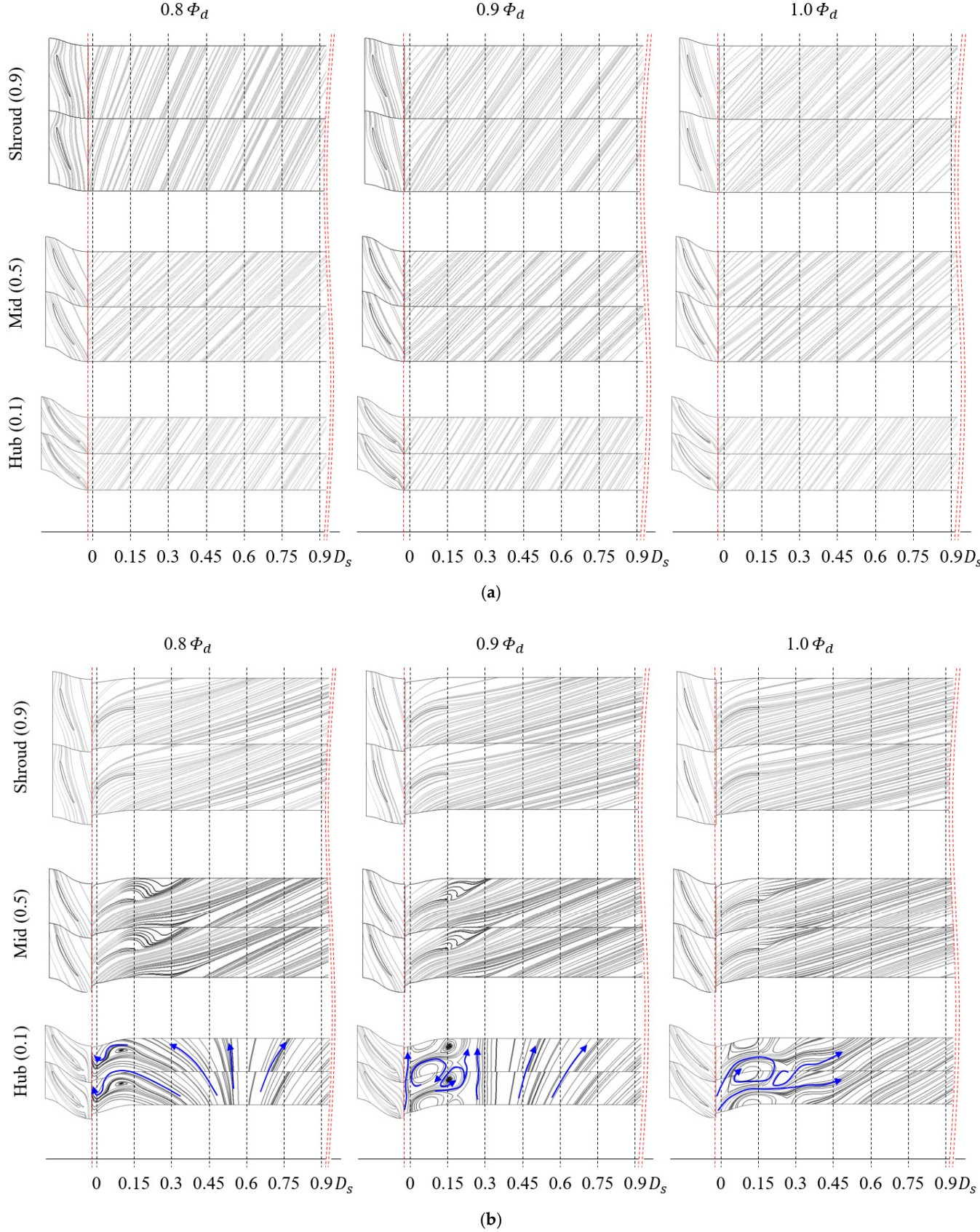

**Figure 13.** *Cont.*

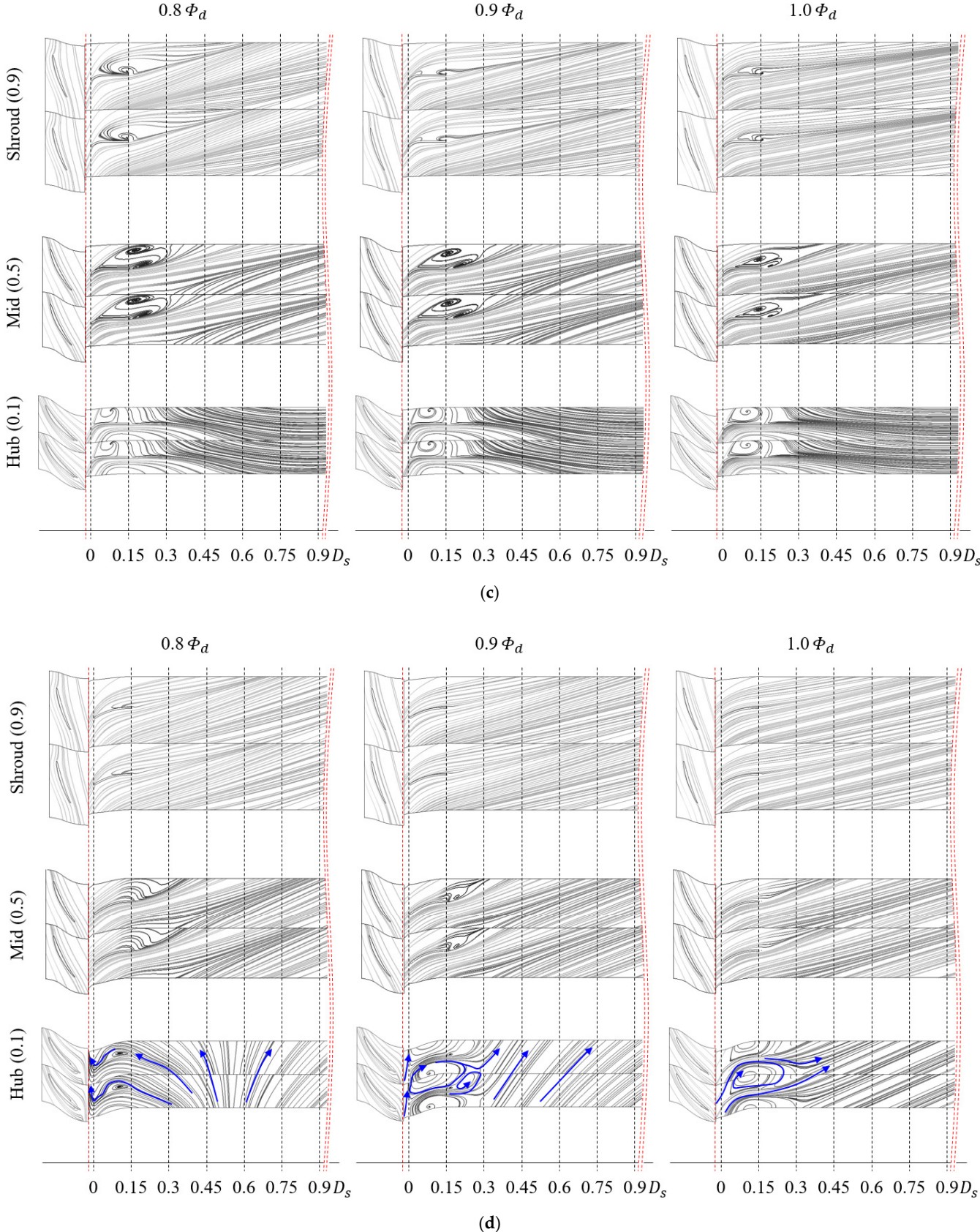

**Figure 13.** *Cont.*

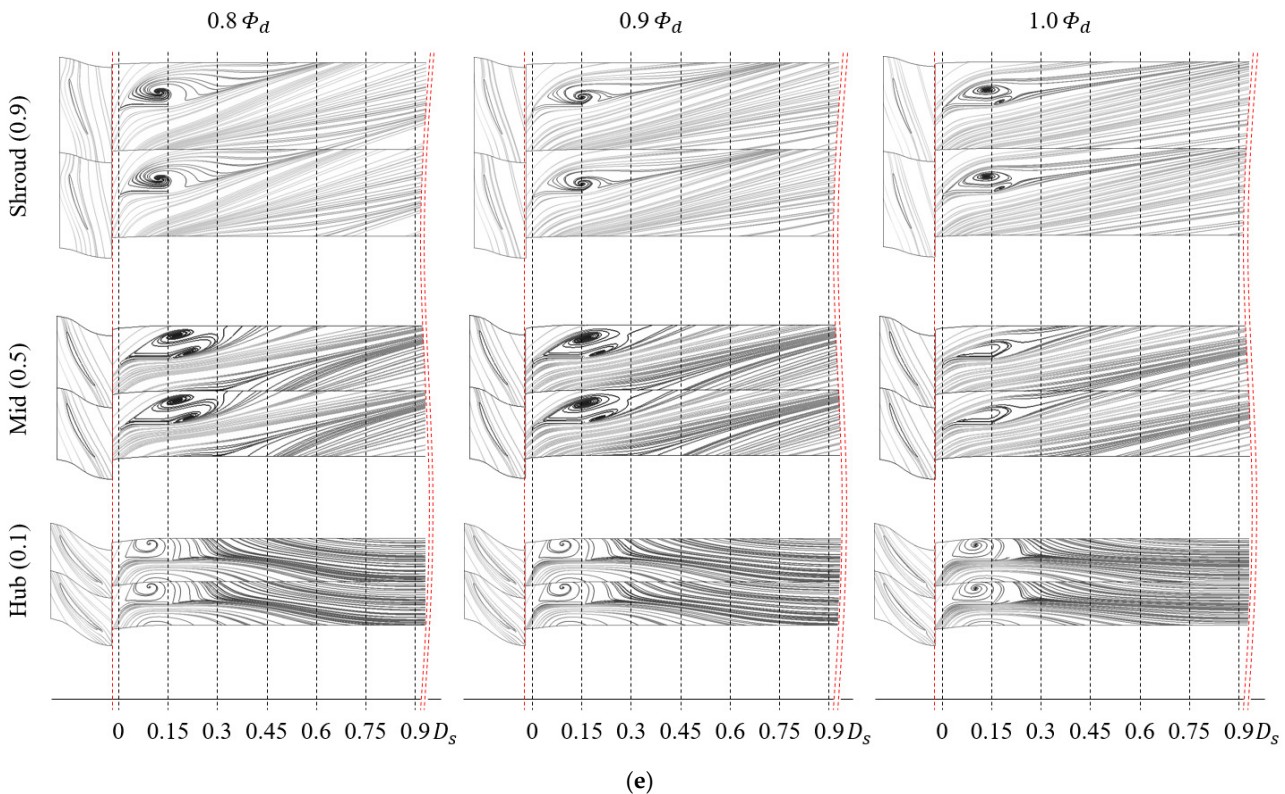

(**e**)

**Figure 13.** Internal flow fields with limiting streamlines in blade-to-blade view: (**a**) none; (**b**) 2D-10; (**c**) 2D-70; (**d**) 3D-10; (**e**) 3D-70.

### 5.2. Further Design

To further complement the previous designs, an additional design was performed, as shown in Figure 14. The previous designs led to the following problems: separation and pressure drop at the guide vane outlet due to a short meridional length; and strong recirculation near the hub span. Thus, the normalized meridional length based on the fan shroud diameter ($D_s$) was extended to 0.25 and denoted with an asterisk (*). The inlet vane angle was also weighted while considering the flow patterns near the hub span and denoted with an apostrophe (').

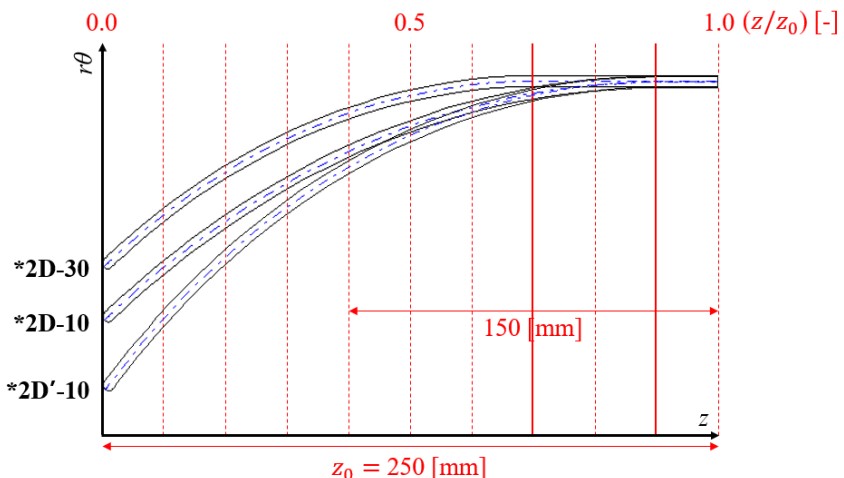

**Figure 14.** Further-designed guide vanes in blade-to-blade view: extended meridional length (*); weighted inlet vane angle (').

Figure 15 shows the static pressure rise curves under the same footnotes of the graph in Figure 12. At 1.0 $\Phi_d$, all three sets underwent static pressure recovery in different patterns. The static pressure still dropped again at the trailing edge of the guide vane, but the range was significantly reduced compared to that shown in Figure 12. Here, the *2D'-10 set contained the best performance. At 0.8 and 0.9 $\Phi_d$, however, the *2D'-10 set showed the worst pressure rise. While the *2D'-10 set performed better for the flow patterns in the blade outlet passage at the design flow rate with the increase of the inlet vane angle, it had the opposite effects at the low flow rates in which the deviation angle increased. For 0.8–1.0 $\Phi_d$, the *2D-30 set could be selected as the best, showing the most pressure recovery based on the point at about 0.5 $D_s$ that suffered a drastic pressure rise (about 0.25 $D_s$ from the vane outlet). As a result, the meridional length of the guide vane had to be extended sufficiently for stable static pressure recovery. Moreover, averaging the flow angle for the entire span at the design flow rate could ensure a better pressure rise over a more comprehensive flow rate range than weighting the flow angle for a specific span.

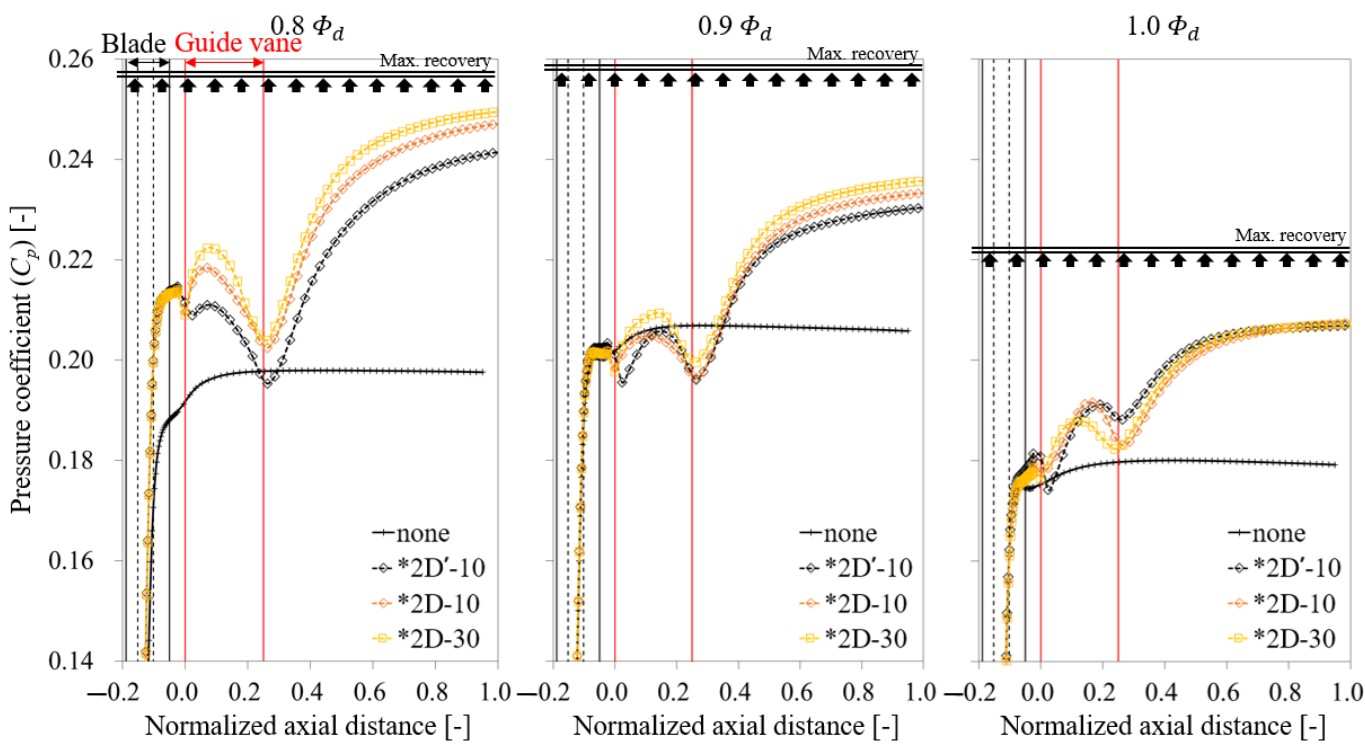

**Figure 15.** Static pressure rise in axial distance.

## 6. Conclusions

In this study, two- and three-dimensional designs for guide vanes with limited meridional length were performed through numerical analysis. The meridional length was as short as that of the blade hub, and the chord length was a variable. The influence of the inlet motor and turbulence model was presented as a previous confirmation. The validation of numerical analysis was also conducted from the experimental test. The recordable results can be summarized as follows:

1.  At design and some low flow rate points, the 2D design contained the most favorable performance when the meridional length (linear) was 30% based on total length, and was the worst for 70%. Here, 'linear' means the vane angle of zero (0) degrees, parallel to the axis with a plate shape.
2.  The 3D design method applied in this study did not outperform the 2D design. In terms of mass production, it is appropriate to consider the 2D design.

3.　If the meridional length of a guide vane is insufficient (short), separation and recirculation can be present near the vane surface, which inhibits the static pressure recovery.

4.　The following solutions can be suggested to achieve better performance with a guide vane: vaneless duct exhibiting length of 0.25 $D_s$ from the vane outlet; and guide vanes with extended meridional length.

5.　In the 2D design concept, averaging the flow angle for the entire span at the design flow rate can ensure a better pressure rise over a more comprehensive flow rate range than weighting the flow angle for a specific span.

6.　The influence of turbulence models between the SST standard and reattachment modification did not have a significant difference in prediction near the design flow rate point.

**Author Contributions:** Conceptualization, Y.-S.C.; methodology, Y.-I.K. and Y.-S.C.; validation, Y.-I.K., Y.-U.C., C.-Y.J., K.-Y.L. and Y.-S.C.; formal analysis, Y.-I.K. and Y.-S.C.; investigation, Y.-I.K., K.-Y.L. and Y.-S.C.; writing—original draft preparation, Y.-I.K. and Y.-S.C.; writing—review and editing, Y.-I.K. and Y.-S.C.; software, K.-Y.L.; resources, C.-Y.J. and K.-Y.L.; data curation, Y.-U.C. and Y.-I.K.; visualization, Y.-I.K.; supervision, Y.-S.C.; project administration, C.-Y.J. and K.-Y.L.; funding acquisition, Y.-S.C. All authors have read and agreed to the published version of the manuscript.

**Funding:** This work was supported by Korea Institute of Energy Technology Evaluation and Planning (KETEP) grant funded by the Korean government (MOTIE) (No. 2021202080026D, Development of platform technology and operation management system for design and operating condition diagnosis of fluid machinery with variable devices based on AI/ICT).

**Institutional Review Board Statement:** Not applicable.

**Informed Consent Statement:** Not applicable.

**Data Availability Statement:** Not applicable.

**Conflicts of Interest:** The authors declare no conflict of interest.

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
