# Peer review of "Effect of Two- and Three-Dimensionally Designed Guide Vanes with Different Camber Length on Static Pressure Recovery of a Wall-Mounted Axial Fan"

_processes, doi:10.3390/pr9091595_

Round 1
Reviewer 1 Report
Dear authors, please refer to the pdf for the comments on the manuscript.

Reviewer 2 Report
Thank you for submitting this paper where you showed a great engineering effort on improving a fan design.
I believe the paper would be of more interest if they include more guidelines for the readers. Right now this example is only analyzing a configuration of 10 blades at 880 rpm and many other parameters which are fixed. There is no indication on how to proceed for a different fan configuration. Furthermore, the geometry is not completely described and therefore, it is not possible to replicate such simulations.
Round 2
Reviewer 1 Report
Thank you for the answer. The paper was improved following the reviewers' suggestions.
Reviewer 2 Report
Thank you for your reply to my comments. I still think the paper is useless for scintific comunity as it does not provide any guidelines for different fan configurations. However, if editor thinks that an engineering job is enough for publication in Processes I think now the work is described with sufficient details.